# A Novel *CARMIL2* Immunodeficiency Identified in a Subset of Cavalier King Charles Spaniels with *Pneumocystis* and *Bordetella* Pneumonia

**DOI:** 10.3390/jof10030198

**Published:** 2024-03-05

**Authors:** Emily L. Coffey, Liang Ma, Ousmane H. Cissé, Joseph A. Kovacs, Katie M. Minor, Antti Sukura, Patrizia Danesi, Steven G. Friedenberg, Jonah N. Cullen, Christiane Weissenbacher-Lang, Julie C. Nadeau, Amber M. Graham, Martin N. Granick, Natalie K. Branson, Kyle C. Branson, Barbara Blasi, Casandra M. Jacobs, Eva Furrow

**Affiliations:** 1Department of Veterinary Clinical Sciences, College of Veterinary Medicine, University of Minnesota, Saint Paul, MN 55108, USA; minork@umn.edu (K.M.M.); fried255@umn.edu (S.G.F.); cull0084@umn.edu (J.N.C.); furro004@umn.edu (E.F.); 2Critical Care Medicine Department, NIH Clinical Center, National Institutes of Health, Bethesda, MD 20892, USA; mal3@cc.nih.gov (L.M.); ousmane.cisse@nih.gov (O.H.C.); jkovacs@mail.nih.gov (J.A.K.); 3Department of Veterinary Biosciences, University of Helsinki, 00014 Helsinki, Finland; antti.sukura@helsinki.fi; 4Laboratory of Parasitology, Mycology and Medical Enthomology, Istituto Zooprofilattico Sperimentale delle Venezie, 35020 Legnaro, PD, Italy; pdanesi74@gmail.com; 5Department of Biological Sciences and Pathobiology, Institute of Pathology, University of Veterinary Medicine Vienna, Veterinärplatz 1, 1210 Vienna, Austria; christiane.weissenbacher-lang@vetmeduni.ac.at (C.W.-L.); barbara.blasi@vetmeduni.ac.at (B.B.); 6Veterinary Specialty Services, Manchester, MO 63021, USA; drjnadeau@vssstl.com (J.C.N.); dragraham@vssstl.com (A.M.G.); 7Pacific Northwest Pet ER & Specialty Center, Vancouver, WA 98686, USA; martin.granick@gmail.com; 8Kirkwood Animal Hospital, Kirkwood, MO 63122, USA; natalie.liberman@att.net; 9St. Luke’s Hospital, Chesterfield, MO 63017, USA; kbransondo@yahoo.com; 10Desert Veterinary Medical Specialists, Peoria, AZ 85345, USA

**Keywords:** CARMIL2, Pneumocystis, dog, canine, primary immunodeficiency, combined immunodeficiency

## Abstract

Pet dogs are a valuable natural animal model for studying relationships between primary immunodeficiencies and susceptibility to *Pneumocystis* and other opportunistic respiratory pathogens. Certain breeds, such as the Cavalier King Charles Spaniel, are over-represented for *Pneumocystis* pneumonia (PCP), suggesting the presence of a primary immunodeficiency in the breed. Here, we report the discovery of a *CARMIL2* nonsense variant in three Cavalier King Charles Spaniel dogs with either PCP (n = 2) or refractory *Bordetella* pneumonia (n = 1). *CARMIL2* encodes a protein that plays critical roles in T-cell activation and other aspects of immune function. Deleterious *CARMIL2* variants have recently been reported in human patients with PCP and other recurrent pneumonias. In addition to opportunistic respiratory infection, the affected dogs also exhibited other clinical manifestations of CARMIL2 deficiencies that have been reported in humans, including early-onset gastrointestinal disease, allergic skin disease, mucocutaneous lesions, abscesses, autoimmune disorders, and gastrointestinal parasitism. This discovery highlights the potential utility of a natural canine model in identifying and studying primary immunodeficiencies in patients affected by PCP.

## 1. Introduction

*Pneumocystis* is a highly diverse genus of yeast-like fungal organisms that inhabit the pulmonary alveoli of mammalian hosts [1,2]. In individuals with underlying immune dysfunction, these organisms can lead to severe opportunistic infections, including life-threatening *Pneumocystis* pneumonia (PCP) [3]. Acquired immunodeficiencies increase susceptibility to PCP in several mammalian species. In humans, common acquired immunodeficiencies associated with PCP include co-morbidities (e.g., human immunodeficiency virus) or iatrogenic causes (e.g., immunosuppressive medications) [3,4]. Primary immunodeficiencies (PIs) also predispose to *Pneumocystis* infections [5]. Characterization of PIs that underlie PCP can help identify at-risk patients and advance disease management strategies for patients with this disease [5].

Pet dogs spontaneously infected with *Pneumocystis canis* are a valuable natural model for studying the role of host immunity and susceptibility to *Pneumocystis* infections [6]. Both Miniature Dachshunds and Cavalier King Charles Spaniels (CKCSs) are disproportionately affected by PCP, suggesting that inherited risk factors contribute to PCP susceptibility in these dog breeds [7,8,9]. In humans, genetic heterogeneity and the high incidence of PCP secondary to acquired immunodeficiencies complicate the discovery of genetic risk factors. In contrast, dog breeds have lower genetic diversity, allowing disease-causing variants to be more readily identified [10]. Dogs also represent an important large animal model for defining host immune responses to pathogens given similarities in the development and function of the immune systems between dogs and humans [11,12].

The purpose of this study was to describe the discovery and clinical presentation of a PI attributed to a *CARMIL2* nonsense variant in two CKCSs affected by *Pneumocystis canis* pneumonia and one CKCS with refractory pneumonia caused by the respiratory pathogen *Bordetella bronchiseptica*. This discovery demonstrates the advantages of using a natural canine model to decipher the dynamics between inherited immunodeficiencies, immune function, and host susceptibility to *Pneumocystis* and other opportunistic respiratory pathogens.

## 2. Materials and Methods

### 2.1. CARMIL2 Variant Discovery

Prior to this study, whole genome sequencing (Illumina HiSeq2500 platform with 150-base paired-end reads) was performed on a respiratory sample obtained from a CKCS previously diagnosed with PCP (dog 1; D1) [13,14]. Quality control of reads, followed by mapping and variant calling, was performed using a previously described pipeline with minor modifications [15,16]. Briefly, the reads were mapped to the dog CanFam3.1 reference genome using BWA version 0.7.15-r1140 [17,18]. GATK tools were used to call single-nucleotide polymorphisms (SNPs) and indels [19]. Variants were filtered to remove those with a Phred-scaled quality score <20 and those with only a single read supporting the variant. Ensembl Variant Effect Predictor (VEP; RRID SCR_007931, v98) was used to annotate variants [20], and variants were further filtered for those predicted to have moderate or high impact on the protein (e.g., splice site, nonsense, frameshift, or missense). Based on the suspected autosomal recessive inheritance of PCP susceptibility in the CKCS breed [21], variants present in a homozygous state in D1 were prioritized. These variants were screened against an internal database of canine whole genome sequencing variant calls for 580 dogs of 57 breeds, including 21 CKCS dogs, to identify private and rare variants (allele frequency <0.005) [16]. For missense variants, pathogenicity was predicted using MutPred2 (RRID SCR_010778) and PolyPhen-2 HumVar (RRID SCR_013189), but variants were not filtered based on these scores [22,23]. The Human Phenotype Ontology database (RRID SCR_006016) was searched for the terms “recurrent respiratory infections” (HP:0002205) and “recurrent pneumonia” (HP:0006532) to determine whether the remaining variants were associated with these phenotypes [24]. A single variant, harboring a nonsense nucleotide change in *CARMIL2*, met this criterion and was selected for further investigation. The Ensembl Variation database was used to determine if the variant is a known SNP in dogs or humans [25,26].

### 2.2. Phenotype Cohorts

After identification of the *CARMIL2* variant in D1, genotyping was performed to screen additional dogs from three phenotypic cohorts. The first cohort consisted of 11 CKCSs with a suspected PI based on a diagnosis of PCP or recurrent pneumonia from other respiratory pathogens. DNA from nine of these dogs was submitted to the University of Minnesota Canine Genetics Laboratory (UMN CGL) for genetic testing for a PI, while DNA from the other two dogs came from a collection at the National Institutes of Health (NIH) from previous research on PCP in dogs [13]. The second cohort included 15 dogs of other (not CKCSs) or unknown breeds that were previously diagnosed with PCP (Appendix A). These samples also came from the collection at NIH, originating from Italy and Austria in previous studies [6,27]. The final cohort included 115 control CKCSs with no PCP or clinical suspicion of a PI. These samples were biobanked at the UMN CGL from previous studies.

### 2.3. CARMIL2 Genotyping

DNA was extracted from either bronchoalveolar lavage (BAL) fluid, whole blood, or cheek swabs during routine veterinary care with informed owners’ consent. To screen for the *CARMIL2* variant in the first and second cohorts, standard PCR was performed to amplify a 406 bp fragment encompassing exon 11 of *CARMIL2* (the location of the nonsense mutation) using forward primer 5′-CTCCAGGTGTCTGCTGAGAG-3′ and reverse primer 5′-TGGTTCTCAGAGCCTTGAGG-3′ on a Bio-Rad T100 thermal cycler (Bio-Rad Laboratories, Inc., Hercules, CA, USA). The thermocycling program started with an initial denaturation at 94 °C for 15 min, followed by 40 cycles with denaturation at 94 °C for 30 s, annealing at 58 °C for 30 s, and extension at 72 °C for 30 s and by a final elongation at 72 °C for 10 min. PCR products were then directly sequenced by Sanger sequencing.

To examine the presence of the *CARMIL2* nonsense variant in the third cohort of control CKCSs, a custom TaqMan SNP Genotyping Assay (Assay ID: ANYM6AW, Thermo Fisher Scientific Inc., Waltham, MA, USA) was performed on a Bio-Rad CFX96 real-time PCR system (Bio-Rad Laboratories, Inc., Hercules, CA, USA). All dogs with PCP in the first and second cohorts were also genotyped for a *CD40L* p.R120* variant initially identified in a Shih Tzu dog with PCP using a previously described PCR and sequencing protocol [28].

Available medical records for dogs that were homozygous for the *CARMIL2* variant were reviewed, and data related to the patient’s clinical presentation, medical history, diagnostic results, medications, and clinical outcomes were collected.

## 3. Results

### 3.1. Whole Canine Genome Sequencing and Variant Filtering

Whole genome sequencing of D1 identified eight variants that passed all filtering steps and were selected for further evaluation. These variants included seven missense variants and a nonsense variant (Table 1); all variants were detected in a homozygous state in D1, as per one of the filtering criteria. None of the missense variants were predicted to be pathogenic by PolyPhen-2 HumVar or MutPred2 (Table 1). One variant was predicted to be “possibly damaging” by PolyPhen-2 HumVar but resided in a gene that has since been retired by Ensembl. A nonsense variant in exon 11 of *CARMIL2* (CanFam3.1 CFA5 g.81801920G>A, Dog10K_Boxer_Tasha CFA5 g.81791628, UniProt A0A8I3NGS3 p.R291*) was identified as the likely pathogenic variant given the role of the gene in immune function and previous reports in humans potentially linking *CARMIL2* variants with PCP [29,30]. *CARMIL2* variants reported to cause PI in human patients result in significantly reduced to undetectable levels of protein expression [29,30,31]. The canine CARMIL2 protein consists of 1394 amino acids; the p.R291* variant identified here results in a loss of 79% of the protein sequence. This variant is a reported SNP in dogs (rs3330142729), with a homologous counterpart also documented in humans (rs1270039585) [26]. Among the 207 dog genotypes listed in the Ensembl Variation database (as of 28 December 2023), only 1 is heterozygous for the p.R291* variant and none are homozygous, with an overall allele frequency of 0.2%; breed information is not reported for these dogs. No phenotype data were reported for the canine SNP. For humans, this site is multiallelic, including both nonsynonymous variants and synonymous SNPs. The nonsense variant (UniProt Q6F5E8 p.R290*) is rare with an allele frequency of 4.2 × 10^−6^ and 1.3 × 10^−5^ in the gnomAD exomes r2.1.1 and gnomAD genomes v3.1.2, respectively; its clinical significance in ClinVar (RRID SCR_006169) was reported as pathogenic by a single submitter based on the known pathogenicity of other *CARMIL2* nonsense variants (no direct evidence provided).

### 3.2. CARMIL2 Genotyping

Genotyping results for *CARMIL2* p.R291*are summarized in Table 2. Of the 11 additional CKCS dogs with clinical suspicion for a PI, 2 were homozygous for the *CARMIL2* p.R291* variant, including 1 with PCP and 1 with refractory *Bordetella* pneumonia. The 9 CKCS dogs and the 15 dogs of other or unknown breeds (Appendix A) with a suspected PI were homozygous for the reference allele. The *CD40L* p.R120* variant previously found in a Shih Tzu dog with PCP was not detected in D1 or the other 26 dogs with clinical suspicion for a PI.

Of the 21 CKCS dogs in the internal database of whole genome sequencing variant calls, 1 was heterozygous, and 20 were clear of the *CARMIL2* variant. Of the 115 control CKCS dogs genotyped by the UMN CGL, 12 were heterozygous and 103 were clear of the variant. The variant allele frequency in the combined group of CKCS database and control dogs was 4.8% (95% CI 2.8–8.0%). The 559 dogs of other breeds in the internal database were all homozygous for the reference allele.

### 3.3. Clinical Summary for CKCSs Homozygous for the CARMIL2 Variant Allele

Clinical data for the three dogs homozygous for the *CARMIL2* p.R291* variant are summarized in Table 3. D1 was an intact male CKCS diagnosed with PCP at 1.5 years of age [14]. The diagnosis was confirmed by cytology, immunohistochemistry with anti-*Pneumocystis* antibody, and PCR using lower respiratory samples (Figure 1). Whole genome sequencing showed that D1 was infected with two *Pneumocystis* populations, denoted as CK1 and CK2, potentially representing distinct species [13]. Serum electrophoresis demonstrated a low gamma globulin fraction and a peak in the alpha-2-globulin fraction. Therapy with trimethoprim–sulfonamide (TMS, 60 mg/kg/day) was initiated, but the patient died the day of starting therapy. Other previous medical history included an abscess in the stifle region, gastroenteritis, a tibial fracture, and erosive oral inflammation, all occurring within the first year of life. Postmortem examinations revealed shrunken lymph nodes and spleen with aberrant histological structures, suggesting lymphatic hypoplasia. 

Dog 2 (D2) was a neutered male CKCS diagnosed with PCP at 16 months of age via PCR testing of a deep oropharyngeal swab. Immunoglobulin assays showed all values within the normal reference intervals (IgM: 350 mg/dL, RI: 100–400 mg/dL; IgG: 1200 mg/dL, RI: 670–1650 mg/dL; IgA: 240 mg/dL, RI: 35–270 mg/dL). Treatment with a 6-week course of TMS (30 mg/kg every 8 h) and an anti-inflammatory dose of prednisone (0.5 mg/kg/day) resulted in resolution of both clinical and radiographic manifestations. After completing TMS therapy, no prophylaxis to prevent recurrence was given. At the time of writing, one year after PCP diagnosis, the patient has not had any relapses of respiratory infection. Additional medical history included chronic diarrhea, atopic dermatitis, an anal gland abscess, and gastrointestinal (GI) parasitism.

Dog 3 (D3) was an intact male CKCS that suffered from a chronic cough since approximately 2 months of age. He was diagnosed with pneumonia secondary to *Bordetella bronchiseptica* and *Mycoplasma cynos* infections at 5 months of age. The diagnosis was made by a PCR panel, cytology, and cultures of endotracheal wash fluid. *Pneumocystis* PCR of the fluid was negative. Prednisone (1 mg/kg/day) and doxycycline therapy (5 mg/kg every 12 h) provided an initial clinical improvement in cough and respiratory effort, though the effects were transient. Bronchoscopy and BAL were performed, and the patient was diagnosed with persistent *Bordetella* infection via culture of the lavage fluid. The patient was then given gentamicin nebulization and enrofloxacin (10 mg/kg/day). At the time of writing, four months after initiating therapy, the patient was still receiving inhaled gentamicin. His respiratory signs have improved overall, though he still experiences an intermittent cough and exercise intolerance. Additional medical history included chronic hypocholesterolemia with mixed bowel diarrhea, giardiasis, coccidiosis, coprophagia, and masticatory muscle myositis (MMM), which was diagnosed at 3 months of age. No other causes of acquired immunodeficiency (e.g., immunosuppressive medication) were noted for any of the three dogs.

### 3.4. Clinical Summary for CKCSs with Suspected PI Clear of the CARMIL2 Variant Allele

Nine CKCSs with suspected PI (six males and three females) tested clear of the *CARMIL2* p.R291* variant. The median age was four months (range: three months–six years), with all dogs less than one year old except for one. All dogs had a history of respiratory signs that were characterized as chronic, recurrent, or refractory to therapy. Respiratory pathogens identified included *Pneumocystis* alone in two dogs, *Bordetella* alone in one dog, and co-infections of *Pneumocystis* and *Bordetella* in one dog, *Bordetella* and *Mycoplasma* in three dogs, and *Bordetella* and *Pasteurella* in one dog. No other causes of acquired immunodeficiency were identified for any dogs.

## 4. Discussion

This report describes the discovery of a novel *CARMIL2* nonsense variant in two CKCSs with PCP and in a third CKCS with refractory *Bordetella* pneumonia. CKCSs are over-represented for PCP when compared to other dog breeds [7,8], suggesting that a PI exists in the breed and increases susceptibility to opportunistic *Pneumocystis* infection. CARMIL2 deficiencies are also a potential underlying risk factor for PCP and other recurrent pneumonia in humans [29,30,31]. In addition to respiratory infection, several clinical parallels have been observed between these dogs and the reported human cases of CARMIL2 deficiencies, including chronic diarrhea, allergic skin disease, mucocutaneous lesions, autoimmune disorders, GI parasitism, and abscess formation [31]. Eighteen other dogs with PCP, including three CKCSs, tested clear of the *CARMIL2* variant (homozygous for the reference allele); these dogs also tested clear for a *CD40L* variant previously reported in a Shih Tzu with PCP [28]. This suggests that additional PIs likely exist in dogs with PCP, including within the CKCS breed. The discovery of the *CARMIL2* variant in dogs with PCP demonstrates shared immune responses that contribute to host defense against *Pneumocystis* across mammalian species. Affected dogs can serve as a natural model to improve our understanding of CARMIL2 deficiencies in humans.

*P. jirovecii* pneumonia was recently described in two young human patients with CARMIL2 deficiencies, although interpretation of this finding is complicated by alternative risk factors present in these patients (e.g., infliximab therapy and stem cell transplant) [29,30]. However, recurrent respiratory tract infection has also been reported as one of the most common infectious manifestations of CARMIL2 deficiencies [31,32]. CARMIL2 deficiencies produce a combined immunodeficiency with defects in T-cell, B-cell, and NK cell function [31,32,33]. CARMIL2 is a multidomain protein expressed in the cytoplasm, particularly of skin, lymphoid tissue, and the GI tract [34]. The protein is necessary for maintaining proper immune function at the cellular and molecular levels due to its roles in actin assembly, cell migration, and T-cell activation. T-cell function relies on both T-cell antigen receptors and co-signaling pathways, such as CD28. The CARMIL2 protein binds to cell membranes through vimentin and ligates CD28 and CARMA1, thereby mediating NF-kB signaling [31,33,34] and promoting T-cell activation. In mice expressing mutated *CARMIL2* genes, CD28-mediated activation of T-cells is impaired, producing downstream detrimental effects on the development of effector memory CD4+ T-cells and regulatory T-cells [35,36].

Although the specific roles and relative importance of different lymphocyte subsets in *Pneumocystis* clearance are not fully understood, total CD4+ T-cells are a critical and well-established component of *Pneumocystis* immunity [37]. The effect of CARMIL2 on T-cell activation, proliferation, and differentiation provides biological support for increased susceptibility to *Pneumocystis* in patients with *CARMIL2* immunodeficiencies. Humans with *CARMIL2* variants also exhibit defects in memory B-cell formation, impaired antibody responses, and alterations in cytotoxic function [31,33,38]. While less is known about the role of B-cells and NK cells in *Pneumocystis* immunity, evidence suggests that these cell types also play important roles [37]. For instance, B-cells are increasingly recognized as having critical roles in antigen presentation during the immune response to *Pneumocystis* [39,40], providing additional avenues by which CARMIL2 deficiencies may increase risk of PCP. Collectively, CARMIL2 deficiencies lead to immune dysregulation that increases susceptibility to opportunistic pathogens like *Pneumocystis*.

Less than 100 human case reports of *CARMIL2* immunodeficiencies have been documented [32], leaving our understanding of *CARMIL2* variants and their biological implications incomplete. As dogs are affected by both the *CARMIL2* variant and *Pneumocystis* infections, they represent a valuable natural model for studying the impact of this PI and host immune responses to *Pneumocystis*. Unfortunately, we are unable to draw conclusions about the interplay between the *CARMIL2* variant and individual *Pneumocystis* species, as genome sequencing and strain identification of the *Pneumocystis* organism was available for only one dog (D1) [13]. No samples were available from other infected dogs for additional *Pneumocystis* genetic testing. Though the causative agent of human PCP (*P. jirovecii*) is phylogenetically different from that of dog PCP (*P. canis*) [13], the clinical parallels in CARMIL2 deficiencies between humans and these dogs indicate important similarities in host immune responses, regardless of the infecting *Pneumocystis* species.

Beyond respiratory infections, other common clinical manifestations in humans with CARMIL2 deficiencies include dermal disease, failure to thrive, other recurrent infections, allergic symptoms, chronic diarrhea, Epstein–Barr virus-related leiomyomas, mucocutaneous candidiasis, skin abscesses, and respiratory allergies [31]. The CKCSs with the *CARMIL2* nonsense variant reported here also exhibited several of these clinical features, including chronic diarrhea, allergic skin disease, skin abscesses, and oral mucocutaneous lesions, with each reported in one or more affected dogs. Impaired regulatory T-cell development can increase the risk of autoimmune disease in human patients with CARMIL2 deficiencies [31,33], and one CKCS also suffered from MMM, an inflammatory myopathy that occurs when autoantibodies against 2M myofibers are produced. Interestingly, other cases of CKCSs developing MMM at a young age have been reported [41,42], but the relationship between this disorder and the *CARMIL2* variant remains unclear. Parasitic infection of the GI tract, such as giardiasis, has also been reported in humans [33], and both giardiasis and other GI parasitic infections were reported in two of the three dogs in this study. One function of CD28 is the promotion of T helper 2 (Th2) cell differentiation, which is a major effector cell for helminth immunity. Therefore, both biological and clinical evidence suggests that human and canine patients affected by the *CARMIL2* immunodeficiency might also be at increased risk for GI parasitism. Collectively, these findings suggest that dogs with CARMIL2 deficiencies exhibit similar clinical manifestations to those observed in humans.

These parallels between canine and human clinical manifestations of *CARMIL2* immunodeficiency support the utility of a canine model for studying the natural course of this PI. Although only three dogs with the *CARMIL2* variant have been identified, we anticipate that greater awareness of this PI among veterinarians, dog owners, and breeders will result in expanded genetic screening efforts and advance our understanding of the *CARMIL2* variant and other immunodeficiencies. Dogs could also be used to study novel therapeutic strategies for immunodeficiencies, such as IL-2 administration, which has been proposed for humans with CARMIL2 deficiencies [29]. The *CARMIL2* variant discovery also highlights how the dog might contribute to understanding host defenses against *Pneumocystis*.

In a 2018 meta-analysis, CKCSs represented nearly 40% of all dogs reported with PCP, raising concerns for a PI within the breed [8]. Interestingly, one study found that CKCSs with PCP have lower IgG levels and higher IgM levels than in breed- and age-matched controls [7], which mirrors the most common immunoglobulin derangements reported in human patients with CARMIL2 deficiency [29,30,31,38]. However, the relationship between CARMIL2 activity and immunoglobulin levels is not fully understood, and only two of the CKCSs reported here had immunoglobulin assays performed. In D1, serum electrophoresis showed a low gamma globulin fraction, whereas in D2, all immunoglobulin levels were within the normal range, though IgM levels were near the high end of the normal range. This finding raises the possibility that *CARMIL2* variants were not responsible for the altered immunoglobulin levels observed in the previously reported CKCSs with PCP [7]. However, complete immunoglobulin assays in additional dogs with the *CARMIL2* variant and PCP are needed to improve our understanding of these relationships. With improved awareness of the *CARMIL2* variant and expanded genetic screening, additional dogs with this immunodeficiency can be identified, thereby offering opportunities for further investigation of CARMIL2 deficiency in both humans and animals.

The absence of the *CARMIL2* variant in the 3 CKCSs and 15 other or unknown breed dogs with confirmed PCP suggests that additional PIs or acquired immunodeficiencies likely contribute to PCP susceptibility in dogs, much as in humans. All dogs with PCP also lacked the previously reported *CD40L* nonsense variant [28]. It is not uncommon for disease-causing variants to be restricted to a single dog breed, as some likely arose after breed formation or became more common in a particular breed due to genetic drift. For example, two different *IL2RG* variants have been reported to cause an X-linked severe combined immunodeficiency in dogs (OMIA: 000899-9615), and each has only been detected in a single breed [43,44,45]. Multiple genetic disorders can also co-exist in a single breed and result in similar disease manifestations, such as the multiple genetic forms of polyneuropathy existing within the Leonberger breed [46]. In the case of PCP, there are likely many different PIs across dog breeds that can contribute to susceptibility and at least one additional PI within the CKCS breed specifically. Therefore, each dog with PCP that tests clear of the described *CARMIL2* and *CD40L* variants represents an opportunity to identify novel genetic defects for susceptibility.

Assays to define the effects of the p.R291* variant on CARMIL2 protein structure, quantity, and lymphocyte function were not performed, which is a limitation of this study. In humans, at least seven distinct homozygous variants have been identified, all of which were considered deleterious due to reduced or undetectable CARMIL2 protein levels in CD4+ and CD8+ T-cells [29,30,31]. Given the clinical evidence of an immunodeficiency in each affected dog, our findings strongly support the presence of a deleterious variant. Additionally, this canine variant produced a premature stop codon, which most commonly results in nonsense-mediated decay of mRNA that impedes protein production [47]. The premature stop codon occurs at residue 219 of the predicted 1394 protein coding amino acids. Thus, even if a truncated protein is produced, it is expected to lack approximately 80% of the expected coding sequence, including four of the five functional domains [35]. This strongly implies a loss-of-function defect with severe deleterious effects. Future studies to quantify CARMIL2 protein production and delineate the functional implications of this variant on host immunity are warranted.

## 5. Conclusions

This study reports the presence of a combined immunodeficiency caused by a *CARMIL2* variant in three CKCSs with PCP or refractory *Bordetella* pneumonia. These CKCSs also suffered from non-respiratory clinical manifestations that parallel *CARMIL2* immunodeficiencies in humans. Given the rarity of *CARMIL2* immunodeficiencies in humans, data from naturally affected dogs might expand the understanding of biological and clinical outcomes for this condition in both species. Furthermore, genetic testing of CKCSs for the *CARMIL2* variant can aid in clinical diagnosis of a PI and guide breeding practices, thus reducing the frequency of this specific immunodeficiency in the breed. Collectively, these results demonstrate the value of a natural canine model for studying the interplay between PIs, host immune responses, and *Pneumocystis* susceptibility.

## Figures and Tables

**Figure 1 jof-10-00198-f001:**
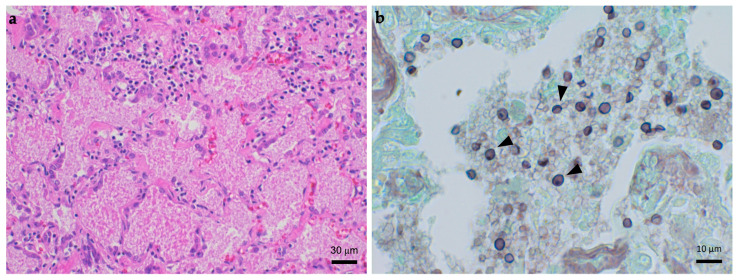
Lung tissue samples from D1 stained with hematoxylin–eosin (HE) and modified Grocott’s methenamine silver (GMS) conducted in a previous study [15]. (**a**) HE staining showing alveolar spaces filled with foamy eosinophilic exudates, a typical characteristic of *Pneumocystis* pneumonia. (**b**) GMS showing *Pneumocystis* cysts (arrowheads) within alveolar spaces.

**Table 1 jof-10-00198-t001:** Genetic variants with moderate-to-high impact (e.g., missense, nonsense, frameshift) identified in a homozygous state in a Cavalier King Charles Spaniel dog with *Pneumocystis* pneumonia and rare in a canine population database.

Genomic Alteration (CanFam3.1)	Gene	UniProt ID	Protein Alteration	PolyPhen-2 HumVar †	MutPred2 †
**CFA5 g.81801920G>A**	** *CARMIL2* **	**A0A8I3NGS3**	**p.R291***	**NA**	**NA**
CFA7 g.39301677C>T	*LBR*	A0A8C0SLC1	p.P356L	0.07	0.29
CFA7 g.40051413G>A	*NVL*	A0A8C0M6L5	p.R134Q	0.00	0.05
CFA14 g.32122502C>G	*ENSCAFG00000002455*	A0A8I3N264	p.R90Q	0.00	0.31
CFA23 g.9564400C>G	*ENSCAFG00000048903* ‡	NA ‡	p.Q618H	0.54	0.47
CFA30 g.38173284A>C	*C30H15orf39*	A0A8C0TAG0	p.E61A	0.12	0.27
CFA30 g.38174844G>A	*C30H15orf39*	A0A8C0TAG0	p.A581T	0.03	0.05

Only the *CARMIL2* variant (in bold) was predicted to be pathogenic for a primary immunodeficiency. † PolyPhen-2 HumVar and MutPred2 scores range from 0–1; a score of ≥0.5 indicates potential pathogenicity, and higher scores have lower false-positive rates. ‡ This gene and its associated transcript (ENSCAFT00000086561) and protein (ENSCAFP00000052210) have been retired.

**Table 2 jof-10-00198-t002:** *CARMIL2* variant (CFA5 g.81801920G>A; p.R291*) allele frequency in dogs with clinical suspicion for a primary immunodeficiency (n = 27) and in population controls (n = 695).

	*CARMIL2* p.R291* Genotypes ^#^			Total Number of Dogs	Variant Allele Frequency (var Alleles/Total Alleles)
	**var/var**	**var/ref**	**ref/ref**		
PI suspects *					
CKCS	3 (2)	0	9 (3)	12	25%
Other/unknown breeds	0	0	15 (15)	15	0%
Breed/population controls					
CKCS	0	13	123	136	4.8%
Other/unknown breed	0	0	559	559	0%

CKCS, Cavalier King Charles Spaniel; PI, primary immunodeficiency; var, variant allele; ref, reference allele. ^#^ The values are the numbers of dogs for each genotype, with the number of dogs with confirmed PCP provided in parentheses. * Includes dogs with confirmed *Pneumocystis* pneumonia (PCP) or recurrent pneumonia from other respiratory pathogens.

**Table 3 jof-10-00198-t003:** Summary of clinical data for three CKCS dogs homozygous for the *CARMIL2* p.R291* variant.

	Age and Sex	Diagnosis	Confirmation of Diagnosis	Other Conditions	Outcome
**D1**	1.5 years, MI	PCP	Cytology and immunocytochemistry (BAL, lung FNA); PCR, immunohistochemistry, electron microscopy (lung tissue, post-mortem)	Skin abscess, early onset GI disease, erosive oral lesions	Died on the day of TMS therapy initiation
**D2**	16 months, MN	PCP	PCR (deep oropharyngeal swab)	Atopic dermatitis, chronic diarrhea, GI parasitism	Complete response to TMS and prednisone
**D3**	5 months, MI	Refractory *Bordetella* pneumonia	Aerobic culture (BAL fluid) and PCR (oropharyngeal swab)	MMM, hypocholesterolemia, chronic diarrhea, GI parasitism	Partial clinical response to inhaled gentamicin

BAL = bronchoalveolar lavage; CKCS = Cavalier King Charles Spaniel; FNA = fine-needle aspirate; GI = gastrointestinal; MI = male intact; MMM = masticatory muscle myositis; MN = male neutered; PCP = *Pneumocystis* pneumonia; TMS = trimethoprim–sulfonamide.

## Data Availability

The whole genome data used in this manuscript are available in NCBI’s Sequence Read Archive with accession number SRR27385252.

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
