# Peer review of "A Novel CARMIL2 Immunodeficiency Identified in a Subset of Cavalier King Charles Spaniels with Pneumocystis and Bordetella Pneumonia"

_jof, 2024, doi:10.3390/jof10030198_

Round 1
Reviewer 1 Report
In this manuscript Emily Coffey et al report CARMIL2 deficiency in Cavalier King Charles Spaniels with Pneumocytis and Bordetella pneumonia, but also early-onset gastrointestinal disease, allergic skin disease, mucocutaneous lesions, abscesses, autoimmune disorders, and gastrointestinal parasitism. CARMIL2 deficiency was reported first in mouse and human in 2013 (Liang Y et al, Nat. Immunol., 2013) and 2016 (Wang Y, J. Exp. Med.), respectively. While the infectious phenotype of CARMIL2 deficiency in inbred mice was unknown because of pathogen free breeding conditions, the human infectious and inflammatory phenotypes have been extensively characterized in large cohorts.
The homozygous variant identified in dog is a premature stop codon early in the coding sequence. Such variant in humans is associated with complete loss of function. I find the report interesting and important as it reveals the phenotype of CARMIL2 deficiency in dog breeds, which is similar to human. Thus, the model may be useful to study CARMIL2 deficiency in a model different from inbred mice living in pathogen free conditions. The paper is well written and easy to read.
One can only regret the lack of functional validation, or at least the proof that the carrier dogs lack CARMIL2 expression. The authors should try the EM53 monoclonal antibody as an easy and quick way to prove CARMIL2 deficiency in the studied animals (by flow cytometry on whole blood or PBMCs, or western blot on PBMCs). The EM53 antibody recognizes CARMIL2 both in mouse and human, and is likely to crossreact with the canine CARMIL2.
None
Author Response
We agree that it would be beneficial to have confirmation that the truncated protein is not expressed. We currently only have access to one living dog that is homozygous for the CARMIL2 variant. Since the EM-53 antibody recognizes an epitope that isn't part of the truncated sequence, it unfortunately could not be used to determine if the truncated protein is expressed or not. However, we could request peripheral blood RNA from the affected dog and design an RT-PCR assay. It would only be for the one case, though we do have multiple controls. Please let us know if you think this would add value or if you know of an antibody that recognizes an epitope in the first 219 residues. We are willing to attempt an expression assay, but we would need some time to try to obtain the appropriate samples and run the assay.
Reviewer 2 Report
The authors describe their discovery of a primary immunodeficiency due to CARMIL2 variant in Cavalier King Charles Spaniels. This is an interesting and innovative work that may lead to better understand the role of this variant in susceptibility to pulmonary infections including PCP.
The scientific content of the manuscript is well described.
Author Response
Thank you for your feedback. We revised the title of this manuscript to “A novel CARMIL2 immunodeficiency identified in a subset of CKCSs with Pneumocystis and Bordetella pneumonia” to capture that only a subset of dogs with PCP or other respiratory pathogens were affected by the variant.
Reviewer 3 Report
This is an excellent report.
No comments
Author Response
Thank you for reviewing our work.